# Biomarker-Based Risk Assessment Strategy for Long COVID: Leveraging Spike Protein and Proinflammatory Mediators to Inform Broader Postinfection Sequelae

**DOI:** 10.3390/v17091215

**Published:** 2025-09-05

**Authors:** Ying-Fei Yang, Min-Pei Ling, Szu-Chieh Chen, Yi-Jun Lin, Shu-Han You, Tien-Hsuan Lu, Chi-Yun Chen, Wei-Min Wang, Si-Yu Chen, I-Hsuan Lai, Huai-An Hsiao, Chung-Min Liao

**Affiliations:** 1Department of Bioenvironmental Systems Engineering, National Taiwan University, Taipei 10617, Taiwan; f00622037@ntu.edu.tw (Y.-F.Y.); d09622004@ntu.edu.tw (W.-M.W.); d13622004@ntu.edu.tw (S.-Y.C.); r12622040@ntu.edu.tw (I.-H.L.); 2Department of Food Science, National Taiwan Ocean University, Keelung City 20224, Taiwan; mpling@mail.ntou.edu.tw (M.-P.L.); 11132002@mail.ntou.edu.tw (H.-A.H.); 3Department of Public Health, Chung Shan Medical University, Taichung 40201, Taiwan; scchen@csmu.edu.tw; 4Department of Family and Community Medicine, Chung Shan Medical University Hospital, Taichung 40201, Taiwan; 5Institute of Food Safety and Health Risk Assessment, National Yang Ming Chia Tung University, Taipei 11221, Taiwan; yijunlin@nycu.edu.tw; 6Institute of Food Safety and Risk Management, National Taiwan Ocean University, Keelung City 20224, Taiwan; shyou@mail.ntou.edu.tw; 7Department of Science Education and Application, National Taichung University of Education, Taichung 403514, Taiwan; thlu@mail.ntcu.edu.tw; 8Department of Environmental and Global Health, College of Public Health and Health Professions, University of Florida, Gainesville, FL 32610, USA; chen.chiyun@ufl.edu; 9Center for Environmental and Human Toxicology, University of Florida, Gainesville, FL 32611, USA

**Keywords:** Long COVID, persistent SARS-CoV-2, symptom number, spike protein, proinflammatory mediators, risk assessment

## Abstract

Long COVID, characterized by persistent symptoms following acute SARS-CoV-2 infection, has emerged as a significant public health challenge with wide-ranging clinical and socioeconomic implications. Developing an effective risk assessment strategy is essential for the early identification and management of individuals susceptible to prolonged symptoms. This study uses a quantitative approach to characterize the dose–response relationships between spike protein concentrations and effects, including Long COVID symptom numbers and the release of proinflammatory mediators. A mathematical model is also developed to describe the time-dependent change in spike protein concentrations post diagnosis in twelve Long COVID patients with a cluster analysis. Based on the spike protein concentration–Long COVID symptom numbers relationship, we estimated a maximum symptom number (~20) that can be used to reflect a persistent predictor. We found that among the crucial biomarkers associated with Long COVID proinflammatory mediator, CXCL8 has the lowest 50% effective dose (0.01 μg mL^−1^), followed by IL-6 (0.39), IL-1β (0.46), and TNF-α (0.56). This work provides a comprehensive risk assessment strategy with dose–response tools and mathematical modeling developed to estimate potential spike protein concentration. Our study suggests persistent Long COVID guidelines for personalized care strategies and could inform public health policies to support early interventions that reduce long-term disability and healthcare burdens with possible other post-infection syndromes.

## 1. Introduction

Long COVID, known as post-acute sequelae of SARS-CoV-2 infection (PASC), is a condition characterized by persistent symptoms that continue for weeks or months after the acute phase of a COVID-19 infection [1,2,3]. Long COVID affects millions of individuals worldwide, with varying degrees of severity and duration [3,4]. A systematic review and meta-analysis highlighted the complexity of Long COVID, noting that symptoms can span multiple systems, including neurological, cardiopulmonary, and gastrointestinal [4]. A recent study also revealed a significant shift in healthcare utilization among Long COVID patients, moving from acute care to outpatient services [5]. This transition reflects the chronic and relapsing nature of Long COVID, which often requires prolonged symptom management rather than emergency intervention. As a result, the condition poses a growing burden on primary and specialty outpatient care, challenging healthcare systems with increased demand for multidisciplinary support. This shift underscores the urgency of developing effective, scalable, and evidence-based treatment strategies that can be delivered in outpatient settings to improve patient outcomes and reduce long-term healthcare costs [6].

Recent studies on Long COVID have provided significant insights into the condition’s complexity and the impact of viral loads on symptoms. Specifically, higher viral loads were associated with more severe respiratory symptoms and restrictive spirometry patterns in recovered individuals [7]. Moreover, slower viral clearance rates during the acute phase, more severe initial symptoms, and underlying chronic conditions or immune dysregulation were linked to an increased risk of Long COVID, particularly among women [8,9,10,11]. Additionally, research has shown that persistent viral presence can lead to prolonged immune activation, contributing to the chronic symptoms observed in Long COVID patients [12].

Importantly, the persistence of SARS-CoV-2 spike protein—even in the absence of active viral replication—has been implicated in the pathogenesis of Long COVID. Circulating spike proteins have been detected in plasma months after initial infection, suggesting it may act as a persistent antigenic stimulus, contributing to ongoing immune activation and chronic inflammation [13,14]. Studies have shown that the spike protein, particularly the S1 subunit, can remain in the body for extended periods, contributing to ongoing immune activation and inflammation [15]. This persistent presence of the spike protein has been linked to various Long COVID symptoms, including fatigue, cognitive dysfunction, and neurological issues [16]. Some patients with Long COVID were found to have SARS-CoV-2 proteins in their blood months after the initial infection, indicating a potential mechanism for the chronic symptoms observed [17]. Additionally, the prolonged detection of spike proteins in plasma or immune cells is found to be associated with sustained inflammation and ongoing symptoms, suggesting it may serve as a predictive biomarker for post-acute sequelae [13,14]. Monitoring spike protein dynamics could therefore enhance the early identification of at-risk individuals and inform targeted intervention strategies [10]. These findings underscore the relevance of spike proteins as a quantitative biomarker for risk assessment and potential therapeutic targeting in Long COVID.

Moreover, research has also highlighted the significant relationship between the SARS-CoV-2 spike protein and the production of proinflammatory cytokines in Long COVID patients [18]. The spike protein has been shown to trigger prolonged expression of cell adhesion markers and the release of various cytokines and chemokines, such as interleukin-6 (IL-6), tumor necrosis factor-alpha (TNF-α), and interleukin-1β (IL-1β), which are commonly observed in severe COVID-19 cases [18]. This persistent inflammatory response is thought to contribute to the chronic symptoms experienced by Long COVID patients. Additionally, the spike protein can activate the NLRP3 inflammasome in macrophages, leading to the release of tissue factor-bearing microvesicles and further promoting inflammation [19]. Another study highlighted a persistent change in blood proteins, suggesting that an overactive immune response might be a contributing factor [20]. In addition, higher levels of leukocytes were found to be associated with more severe symptoms in older women [21]. These findings suggest that the spike protein plays a crucial role in sustaining the inflammatory environment that underlies many of the long-term symptoms associated with Long COVID.

Despite the substantial current findings on investigating the role of the spike protein in influencing Long COVID symptoms and expression of preinflammatory cytokines, its systematic characterization in Long COVID patients using mechanistic approaches remains limited. We thus aimed to explore the role of spike proteins in Long COVID by investigating the dose-dependent effects and temporal profile based on the developed mathematical models. Therefore, the objectives of this study are threefold: (i) characterize the dose–response relationships between spike proteins and Long COVID symptom numbers and the release of proinflammatory mediators, (ii) quantify the time-dependent spike protein concentration in Long COVID patients, and (iii) provide a comprehensive risk assessment strategy using the developed dose–response tools and mathematical modeling to facilitate personalized monitoring and early therapeutic interventions.

## 2. Materials and Methods

### 2.1. Study Data and Framework

To develop risk assessment strategies by mechanistically investigating the role of spike proteins in Long COVID, we adopted three studies to explore both dose-dependent and time-dependent profiles of spike proteins in PASC patients [22,23,24] (Figure 1). For dose-dependent effects, the spike protein concentration–Long COVID symptom numbers and spike protein concentration–expression of proinflammatory mediator relationships were constructed [22,24]. The spike protein dynamic monitoring for individual PASC patients [23] was also mechanistically described with the developed mathematical model. We applied the clinical data from Swank et al. [23], in which plasma samples from a cohort of 63 individuals previously infected with SARS-CoV-2 were analyzed, 37 of whom were diagnosed with post-acute sequelae of SARS-CoV-2 infection (PASC). Among the 37 patients with PASC, only 12 had longitudinal samples. Although there was no control for individuals without SARS-CoV-2 infection, we provided a temporal proofing of individuals not diagnosed with PASC (*n* = 6) for comparison with the time-dependent spike protein concentration in PASC patients (*n* = 12) in Appendix A. The mode of action of spike proteins on Human Lung Macrophages (HLMs) was also discussed [24].

### 2.2. Dose–Response Model

We derived the dose–response relationships for spike protein concentration (log copy μL^−1^) versus Long COVID symptom numbers (Appendix A) [22] and the expression of proinflammatory mediators [24] by fitting the three-parameter Hill model to datasets in the published literature, as follows: $E(D)\;={\;E}_{max}/(1\;+{\;\left(ED50/D\right)}^{n})$. *D* is the spike protein concentration (log copy μL^−1^), *E_max_* is the maximum Long COVID symptom numbers or fraction of protein expression compared to lipopolysaccharide (LPS) at 1 µg mL^−1^, *ED*50 is the dose causing an effect equal to 50% *E_max_* (symptom numbers or fraction of protein expression), and *n* is the fitted Hill coefficient. The Hill model provides a robust and interpretable framework for modeling sigmoidal dose–response relationships, which are commonly observed in biological systems involving receptor–ligand interactions and inflammatory signaling. This model allows us to quantitatively characterize the potency (*ED*50) and steepness (*n*) of spike protein-induced effects, facilitating comparison across datasets and enabling a more mechanistic understanding of dose-dependent symptom manifestation and cytokine activation in Long COVID.

### 2.3. Development of Time-Dependent Modeling of Spike Protein Concentration

To characterize dynamic changes in spike proteins in Long COVID patients, data points of measured spike protein concentrations were extracted and a cluster analysis was applied to group patients with similar temporal patterns of spike protein levels [23]. The optimal number of clusters was determined using a scree plot, which evaluates the within-group sum of squares (WSS) to identify the point at which adding more clusters results in diminishing returns on explained variance. The time-dependent spike protein concentrations were then fitted with a nonlinear statistical model—*D* = a + b*t*^2^ + ce*^t^* + d*t*/ln*t*—where *D* is the spike protein concentration (log copy μL^−1^), *t* is time (month), and a, b, c, and d are fitted coefficients.

### 2.4. Data Analysis

Mathematical model fittings with uncertainties were conducted using the TableCurve 2D (Version 5.01, AISN Software, Mapleton, OR, USA). A cluster analysis was conducted by using the cluster package in R language (Version 4.1.3, The R Foundation for Statistical Computing).

## 3. Results

### 3.1. Effect Analyses of Long COVID Symptom Numbers

To derive the relationship between spike protein concentration (log copy μL^−1^) and Long COVID symptom numbers, the linear relationship between the value of cycle threshold (*C*_t_) and the viral load of SARS-CoV-2, based on Gene E, was first determined (*y* = −3.69*x* + 40.87, *r*^2^ = 0.99, *p* < 0.001; Figure 2a; Appendix A). Subsequently, a three-parameter Hill-based model was also applied to assess the relationships between spike protein concentration (log copy μL^−1^) and Long COVID symptom numbers (Figure 2b; Appendix A). The results showed that the Hill-based model could mechanistically derive the Long COVID symptom numbers based on spike protein contents (*r*^2^ = 0.7, *p* < 0.001) with a mean *E*_max_ of 19.83 and *ED*50 of 3.77 log copy μL^−1^, which is approximately 1.8 × 10^−6^ µg mL^−1^ (assuming spike monomer MW = 180 kDa) (Appendix A).

### 3.2. Time-Dependent Modeling of Spike Protein Concentration

To better reflect the relationships between time and spike protein concentration in twelve Long COVID patients (Appendix A), a cluster analysis was first performed to group similar data points into clusters based on k-means clustering. Five clusters were characterized based on the results of a within-group sum of squares (SS) (Appendix A). The silhouette plot indicates that the five-cluster solution provides moderate overall validity (mean silhouette width = 0.42) (Figure 3a). Most spike protein concentration data have positive coefficients, typically between 0.1 and 0.6, signifying that they are better matched to their assigned cluster than to any other. This suggests heterogeneity in spike protein levels among Long COVID patients, potentially reflecting different underlying biological mechanisms or symptom profiles. Thus, the 48 data points of the twelve Long COVID patients can be characterized by five clusters in the time–spike protein concentration relationship (Figure 3b). The points in clusters 1, 2, and 4 are positioned very close to the convex hull boundaries of neighboring clusters, indicating potential overlap or proximity in feature space. As a result, these boundary observations exhibit negative silhouette widths (Figure 3a). Subsequently, a mathematical model was applied to project the time-dependent spike protein concentrations (Figure 4). We showed that the distributions of the 48 data points under the five clusters were appropriately grouped (Figure 4a). We further determined that the time-dependent spike protein concentration changes in patients were statistically significant (*r*^2^ = 0.99; *p* < 0.001) and conducted a trend analysis (average spike protein concentration in each month) to present a comparison (Figure 4b; Appendix A).

### 3.3. Dose–Response Between Spike Proteins and Proinflammatory Mediator Expression

A conceptual model illustrates how spike proteins induce the release of proinflammatory cytokines (e.g., IL-6, IL-1β, and TNF-α) and chemokine (e.g., CXCL8) from human lung macrophages (HLMs), thereby promoting phagocytosis, which contributes to innate immunity (Figure 5a). This response is followed by the recruitment of various inflammatory cells, including neutrophils, monocytes, and natural killer cells. To describe the relationships between spike protein concentration and the expression of proinflammatory mediators (fraction of protein expression compared to LPS at 1 µg mL^−1^), a three-parameter Hill-based model was applied (Figure 5b–e). Our results indicated that the Hill-based model could appropriately present the nonlinear proinflammatory mediators–spike protein relationships (*r*^2^ = 0.38 − 0.98, *p* = 0.001 − 0.06) (Figure 5b–e; Appendix A). We showed that among the four proinflammatory mediators, IL-1β presented the highest mean *E*_max_, of 0.88, followed by CXCL8 (0.72), IL-6 (0.16), and TNF-α (0.06), whereas CXCL8 had the lowest *ED*50, of 0.01 μg mL^−1^, followed by IL-6 (0.39), IL-1β (0.46), and TNF-α (0.56) (Figure 5b–e; Appendix A).

## 4. Discussion

### 4.1. Spike Proteins in Long COVID Patients and Their Effects

Recent studies have identified the persistence of SARS-CoV-2 spike proteins in the blood of individuals suffering from Long COVID, suggesting it may be a biomarker and potential contributor to prolonged symptoms [25]. A study revealed that spike protein was detectable in a majority of Long COVID patients up to 12 months after their initial infection, despite the absence of active viral replication [23]. These residual spike proteins are hypothesized to evade immune clearance and remain in monocytes or other tissues, serving as a chronic immunogenic stimulus. This persistent antigen presence is strongly associated with systemic inflammation and immune dysregulation, including T cell activation and cytokine production. Furthermore, patients with detectable spike proteins tend to report more severe or prolonged symptoms, including fatigue, cognitive impairment, and cardiovascular irregularities. This finding points toward a mechanism involving viral protein persistence—rather than ongoing infection—as a driver of chronic symptoms. They also proposed that spike proteins may remain due to reservoirs of virus in tissues such as the gastrointestinal tract, releasing viral components intermittently into the bloodstream. Similarly, the temporal changes in any Long COVID symptom also showed a steep decrease initially (from 92% at acute phase to 55% at 1-month follow-up), followed by stabilization at approximately 50% during 1-year follow-up [26].

The presence of spike proteins has important implications for immune system dysregulation. They have been shown to act as a superantigen, triggering excessive immune activation and potentially contributing to autoimmunity and persistent inflammation [27]. This sustained immune response can result in a cascade of symptoms commonly reported in Long COVID, such as fatigue, brain fog, and cardiovascular issues. The interaction of spike proteins with toll-like receptors and other immune pathways may also lead to the production of cytokines and inflammatory mediators, creating a chronic inflammatory state [28]. Notably, studies also observed that patients with early post-acute immune dysregulation—marked by interferon-γ persistence and complement system activation—were more likely to experience sustained high spike levels [15]. These findings underscore the biological possibility that circulating spike proteins play an active role in the development and maintenance of Long COVID pathology, possibly through mechanisms of chronic immune activation, endothelial injury, and localized tissue inflammation.

Furthermore, spike proteins have been associated with the vascular and neurological effects that may underlie many Long COVID manifestations. Evidence suggests that they can disrupt the blood–brain barrier and impair endothelial function, leading to neuroinflammation and cognitive dysfunction [29]. Additionally, studies have indicated that spike proteins may contribute to microclot formation and platelet activation, which have been documented in Long COVID patients and are thought to impair oxygen delivery and tissue perfusion [30]. These pathological mechanisms support the hypothesis that circulating spike proteins are not merely a remnant of infection but active participants in the disease process of Long COVID.

To further explore the clinical relevance of these mechanisms, we quantitatively described the relationship between SARS-CoV-2 viral load concentration and symptom numbers based on the data of COVID-19 syndrome in outpatients from Girón Pérez et al. [22]. While the questionnaire was standardized and administered at uniform post-infection intervals, symptoms can evolve beyond the three-month point and self-reported data may be subject to recall bias. Additionally, comorbidities were not evaluated, and the study only included outpatients with mild to moderate COVID-19; more severe or hospitalized cases were excluded, which could limit generalizability. Future studies should aim for longer follow-up durations, include validated symptom instruments, and control for comorbid conditions and other potential confounders.

In addition, our clustering approach in the time-dependent modeling of spike protein concentration captures state membership over time, meaning that, at each time point, a patient is assigned to a state (or group) that reflects a specific profile of spike protein concentration, rather than tracking a continuous, patient-specific trajectory. Because biomarker profiles can change, individual patients may appear in multiple clusters across different time points. The implications for personalized care lie in recognizing these state-specific biomarker patterns and determining which states are more strongly associated with persistent symptoms or elevated spike protein levels, rather than relying on a single fitted curve for each patient. The observed M-shaped pattern in spike protein concentration reflects two apparent peaks separated by a decline over the longitudinal sampling period. Unfortunately, in the dataset from Swank et al. [23], symptom severity was not recorded at the same frequency or time points as the biomarker measurements, and detailed longitudinal symptom scores were therefore not available for direct, within-patient trend comparison. While this prevents us from confirming whether symptom trajectories mirrored the M-shaped biomarker pattern, the biomarker fluctuations remain valuable for understanding potential biological processes that could underlie symptom variability in long COVID. Importantly, we also observed a dose–response relationship between spike protein concentration and the number of reported post-COVID symptoms at the population level. This relationship provides a basis for correlating specific spike protein concentrations with symptom burden in future studies, which may help characterize patients at higher risk for long COVID and inform targeted monitoring or interventions.

### 4.2. Role of Proinflammatory Mediators in Long COVID

Proinflammatory mediators play a central role in the pathophysiology of Long COVID by perpetuating immune activation and tissue damage long after the resolution of acute infection. Elevated levels of cytokines (e.g., IL-6, TNF-α, and IL-1β) have been consistently reported in patients with persistent symptoms, suggesting a chronic inflammatory state [31]. This prolonged cytokine response may result from immune dysregulation initiated during the acute phase of SARS-CoV-2 infection, including impaired interferon signaling and incomplete resolution of inflammation. These mediators can contribute to systemic symptoms such as fatigue, myalgia, and neurocognitive impairment by affecting the central nervous system, promoting endothelial dysfunction, and interfering with cellular energy metabolism [32].

Several studies have demonstrated that elevated proinflammatory cytokines such as IL-1β, IL-6, and TNF-α are significantly associated with greater symptom severity in Long COVID, supporting the biological relevance of our observed spike protein–cytokine correlations in this study. Wynberg et al. [33] reported that individuals with post-acute sequelae of SARS-CoV-2 infection (PASC) exhibited elevated IL-10, IL-17, IL-6, IP10, and TNF-α at 24 weeks post-infection, and that early IL-1β levels predicted PASC at 24 weeks. Schultheiß et al. [34] identified a cytokine “triad” (IL-1β, IL-6, and TNF-α) strongly associated with Long COVID symptoms in a large digital cohort. It was also implied that the association of persistent symptoms with these cytokines could be a mechanistic contributor to the pathogenesis of Long COVID.

In a symptom–cytokine phenotype study, IL-6 and IL-27 elevations were linked to fatigue, while IL-8 was strongly associated with dyspnea [35]. Additionally, Durstenfeld et al. [36] reported that elevated IL-6 correlates with reduced exercise capacity and chronotropic incompetence in Long COVID patients. Low et al. [37] provided a comprehensive outline of cytokine-mediated pathophysiology in Long COVID, focusing on IL-1β, IL-6, and TNF-α as central players in symptom burden. Robineau et al. [38] also found associations between certain symptoms and biomarkers linked to severity, such as IL-8 and CD163, which play roles in multiorgan dysfunction and infection resolution, suggesting that inflammatory biomarkers may aid in diagnosing PASC in its early phase and in assessing the risk symptom persistence. Taken together, these findings support the clinical significance of our results and suggest that the observed spike protein–cytokine patterns may reflect underlying mechanisms contributing to symptom persistence in Long COVID.

Although our results do not show a direct relationship between proinflammatory cytokines and symptom severity, we present findings on the dose–response relationship between spike protein concentration and proinflammatory cytokines, as well as the relationship between spike protein concentration and symptom number. The models constructed for these relationships can be applied to interpret potential links between cytokine levels and symptom severity, providing a framework for future studies to explore the mechanistic pathways underlying Long COVID. Taken together, these findings support the clinical significance of our results and suggest that the observed spike protein–cytokine patterns may reflect underlying mechanisms contributing to symptom persistence in Long COVID.

In addition to cytokines, other proinflammatory molecules, such as C-reactive protein (CRP), monocyte chemoattractant protein-1 (MCP-1), and vascular endothelial growth factor (VEGF), have been implicated in the persistence of symptoms and tissue remodeling in Long COVID [39]. These biomarkers are often elevated in individuals experiencing cardiovascular and respiratory complications, suggesting ongoing vascular inflammation and damage. Furthermore, the prolonged activation of innate immune cells, including monocytes and macrophages, can sustain this inflammatory environment, creating a feedback loop that prevents immune homeostasis. Understanding the role of these proinflammatory mediators not only clarifies the underlying biology of Long COVID but also opens the door to targeted therapies, such as anti-cytokine treatments or immunomodulators, to mitigate long-term sequelae.

Escalating concentrations of recombinant spike can activate human lung macrophages, triggering the production of proinflammatory cytokines that mirror the “cytokine storm” signature observed in clinical COVID-19 cases (IL-2, IL-4, IL-6, IL-1β, IL-8, IFN-γ, TNF-α, and CXCL10) [40,41]. Our dose–response analysis of the changing chemokine/cytokine levels included monocyte/macrophage (CXCL8), proinflammatory cytokines (IL-6), and cytokines that promote innate and adaptive immune response (IFN-γ and TNF-α). Notably, serum levels of IL-6 and TNF-α serve as significant predictors of disease severity and mortality [31], with IL-6 being particularly abundant in severe cases and potentially mediating neuropsychiatric symptoms in Long COVID [42,43]. Spike-mediated immune-escape mechanisms amplify inflammasome-triggered pyroptosis and damage-associated molecular patterns release, overactivating macrophages and natural killer cells beyond physiological antiviral responses [44].

On the other hand, cytokine profiles differ between acute and Long COVID, with acute cases showing elevated IL-6 levels while Long COVID patients exhibit increased IL-2 and IL-17 [44]. Individuals without persistent symptoms have higher IL-6, IL-10, and IL-4 levels. The injection-induced multisystem inflammation syndrome (MIS) presents inflammatory risks comparable to those triggered by viral infection. Proinflammatory cytokines have been detected in serum following Pfizer/BioNTech BNT162b2 mRNA vaccination, with certain monocyte populations undergoing epigenetic conditioning that results in enhanced cytokine production (particularly IFN-γ, CXCL10, and TNF-α) upon booster vaccination [41]. While both COVID-19 and MIS patients exhibit similar activation of the coagulation cascade and elevated LDH levels, these biomarkers are absent in Long COVID patients, suggesting distinct pathogenic mechanisms [45]. Future research must focus on differentiating between infection-derived and injection-derived spike protein exposure to enable accurate risk assessment and proper attribution of inflammatory outcomes.

Moreover, dysbiotic microbiome-induced proinflammatory microenvironment can promote SARS-CoV-2 entry via angiotensin-converting enzyme receptor-2 (ACE2)**,** serine transmembrane TMPRSS2, and possibly other non-canonical pathways [46]. SARS-CoV-2 infection alters the oral microbiome, deteriorating oral health and playing a role in the development of Long COVID [46]. The dysbiotic microbiome in Long COVID patients leads to decreased short-chain fatty acid-producing bacteria (*Faecalibacterium, Ruminococcus*, *Dorea*, and *Bifidobacterium*), impaired enteroendocrine cell function, and increased gut permeability, promoting the release of proinflammatory cytokines [47]. A recent study also found that the number of platelet–leukocyte aggregates is significantly associated with residual lung damage that sustains the most frequently referred symptoms, such as dyspnea, chest pain, and fatigue at rest and after exertion [48]. Long COVID patients show a unique pattern of platelet activation, along with low-grade inflammation mainly driven by C-reactive protein and IL-6, which is linked to the level of platelet activation.

### 4.3. Implications for Long COVID Risk Assessment

The growing understanding of Long COVID’s complex pathophysiology has significant implications for improving risk assessment frameworks. Risk assessment of Long COVID involves identifying factors that predispose individuals to develop persistent symptoms following acute SARS-CoV-2 infection. Emerging research suggests that a more holistic approach—integrating demographic, clinical, immunological, and psychosocial factors—can better predict who is at risk. For instance, in a large cohort study from the UK, the presence of more than five symptoms during the first week of infection strongly predicted the risk of developing Long COVID [49]. Studies also showed that women, older adults, and individuals with pre-existing conditions, such as asthma or autoimmune diseases, are more likely to experience Long COVID [26,49,50]. Additionally, a high symptom burden in the early phase of COVID-19, particularly fatigue, headache, and shortness of breath, is strongly associated with long-term complications [26]. These findings highlight the need for the early identification and stratification of at-risk individuals based on a multi-dimensional set of predictors.

Immunological and genetic predispositions are increasingly recognized in risk stratification. For example, individuals with delayed clearance of viral RNA and early reactivation of latent viruses such as Epstein–Barr virus were more likely to experience Long COVID symptoms [51]. This suggests that immune system dysregulation may underlie some long-term complications. Low levels of certain immune markers, like interferon-γ and IL-2, during acute infection were associated with poor recovery and symptom persistence [51]. Incorporating biomarkers such as persistent viral antigens, proinflammatory cytokines (e.g., IL-6, CXCL8), and immune cell dysregulation into risk assessment models can significantly enhance their predictive power. Persistent elevations in inflammatory mediators known as IL-1β and TNF-α have been correlated with symptom duration and severity. These immunological indicators, when further integrated with clinical and demographic data, can form the basis of a precision medicine approach to Long COVID risk assessment. Such an approach could facilitate proactive monitoring and tailored treatment to inform public health strategies for managing the long-term impacts of the pandemic. To expand on the predictive utility of immune indicators, our study establishes a mechanistic risk assessment strategy based on spike protein concentrations and multiple inflammatory biomarkers, including CXCL8, IL-1β, IL-6, and TNF-α. By quantifying spike protein levels in patient samples and correlating them with symptom severity and inflammatory profiles, our approach enables the early identification of high-risk individuals and offers a mechanistic basis for interventions aimed at preventing or alleviating Long COVID manifestations.

A critical factor in applying this strategy, however, is the timing of testing, as SARS-CoV-2 spike (S) protein levels may fluctuate over time post-infection. Early detection could help identify individuals at risk for developing persistent symptoms. Studies have shown that detectable IgG-class antibodies against SARS-CoV-2 develop approximately 8 to 11 days following the onset of symptoms [52]. Additionally, research indicates that the spike protein can persist in the brain’s protective layers for up to four years after infection, potentially contributing to long-term neurological effects [53]. These findings suggest that conducting tests within the first few weeks post-infection may be most effective for assessing acute immune responses, while longer-term monitoring could be necessary to evaluate persistent viral components and their implications for Long COVID.

Incorporating spike (S) protein detection into diagnostic protocols, alongside clinical evaluation and other biomarkers, could enhance the accuracy of Long COVID diagnosis. Integrating S protein measurements with clinical data could support predictive models, enabling better risk assessment and the identification of individuals more likely to experience prolonged symptoms. Studies have identified various biomarkers that are relevant for diagnosing and monitoring Long COVID. These include proinflammatory cytokines such as interleukin-6 (IL-6) and tumor necrosis factor-alpha (TNF-α), which are elevated in patients with Long COVID and correlate with disease severity [54]. Additionally, biomarkers reflecting SARS-CoV-2 persistence, such as viral RNA detection and the reactivation of latent viruses, have been associated with Long COVID [55]. Markers of endothelial dysfunction and coagulation, such as soluble urokinase plasminogen activator receptor (suPAR), have also been implicated in Long COVID pathophysiology [56]. Furthermore, noncoding RNAs (ncRNAs) have shown potential as biomarkers for disease stratification in COVID-19, which could extend to Long COVID [57]. These findings underscore the importance of a multi-biomarker approach in the diagnosis and management of Long COVID.

The mathematical model built from pooled longitudinal measurements of spike protein, combined with a cluster analysis that groups datapoints by time and concentration in this study, provides a practical way to reveal temporally distinct antigen persistence phenotypes despite inter-individual heterogeneity. These clustered spike trajectories capture meaningful biological information, as persistent spike patterns co-occur with elevated levels of proinflammatory cytokines (IL-1β, IL-6, and TNF-α) previously linked to PASC severity [33,34], and are associated with immune dysregulation signals such as enhanced neutrophil activity and the altered NK-cell phenotypes observed in Long COVID cohorts [58,59]. By mapping cluster membership and model-derived trajectory features (including peak levels, rate of decline, area under the curve, and persistence at late timepoints) onto clinical outcomes, this framework enables risk stratification and the identification of high-risk phenotypes. Mechanistically informed predictors that integrate spike dynamics with cytokine panels, antibody kinetics, and clinical covariates have been used effectively in within-host and prognostic models and can be estimated using hierarchical mixed-effects or joint-model approaches to account for irregular sampling and patient heterogeneity [37,60]. Importantly, our clustering approach was intentionally designed to capture state-specific biomarker profiles at each time point rather than to follow a continuous, patient-specific trajectory. This means that a patient’s sample may be assigned to different clusters over time, reflecting biologically meaningful variability such as intermittent antigen persistence, immune reactivation, or partial clearance. While this may appear as “jumping” between clusters, we interpret these transitions as indicators of heterogeneous immune responses rather than misclassification. The clinical utility lies in recognizing that certain states (e.g., persistently high spike protein levels with elevated cytokines) are more strongly associated with increased symptom burden and thus represent higher-risk periods for an individual patient. We acknowledge, however, that trajectory-based clustering could complement our approach by capturing entire longitudinal courses, and we note this as a limitation, with the potential to strengthen the study’s potential application to patient-specific care strategies in future work. Together, these studies support the value of the pooled, time-resolved spike and cluster approach as a bridge between mechanistic biomarker behavior and clinically actionable risk scores for Long COVID.

Take Cluster 1 as an example, defined by moderate-to-high spike protein concentrations (3.91–24.04 × 10^−6^ μg mL^−1^) persisting over 4 months (Figure 4a), which likely reflects delayed clearance or intermittent antigen persistence. Patients in this group can be characterized by both biomarker and clinical profiles. Modeled trajectory features for each patient, such as baseline and peak levels, decay slopes, area under the concentration–time curve (AUC), and persistence metrics, provide quantitative summaries that align with elevated proinflammatory cytokines (IL-1β, IL-6, and TNF-α) and other inflammatory markers reported in Long COVID cohorts [33,34,37]. Clinically, these patients may present with fatigue, post-exertional malaise, cognitive complaints, or dyspnea, alongside reduced functional capacity linked to cytokine elevations [59]. To further define risk, recommended follow-up includes cytokine panels, antibody titers, endothelial/coagulation markers, and functional assessments, which, when integrated into the mathematical model, improve the prediction of persistent symptoms [33,58]. For example, a patient in Cluster 1 with a high 4-month AUC and elevated IL-6/IL-1β could be classified by the model as having a ~70% probability of moderate–severe Long COVID at 6 months, prompting referral to a specialized clinic, advanced cardiopulmonary and neurocognitive testing, and prioritization for biomarker-guided therapeutic trials. While this approach demonstrates how cluster-based kinetic modeling can be combined with clinical findings for individualized risk stratification, external validation remains essential to establish reliable thresholds and generalizability [60].

Also, to manage the escalating burden of Long COVID, healthcare systems must adopt integrated, multidisciplinary care models that incorporate validated risk stratification frameworks. Such models could involve primary care providers using standardized screening tools to assess risk profiles—including factors such as symptom burden during acute infection, comorbidities, sex, age, and biomarker levels—to determine the need for referral to post-COVID clinics or specialist care [61,62]. Furthermore, a comprehensive mechanistic model incorporating antibody pharmacokinetics and deep mutational scanning and regional genomic surveillance data could compute how much, and for how long, recovery from a recent infection with a SARS-CoV-2 variant will protect against another variant, enabling the prediction of future variant dynamics and informing risk assessments of variants and vaccine design [63]. This proactive approach would improve clinical outcomes by enabling earlier diagnosis and intervention, reducing unnecessary referrals and service bottlenecks. Furthermore, centralized data systems linking risk profiles with longitudinal outcomes can help refine predictive models, inform dynamic resource planning, and guide research priorities.

In addition to improving patient-level care, risk assessment also has broader socioeconomic implications. By predicting who is most likely to experience long-term disability, health systems can anticipate impacts on workforce participation, disability claims, and mental health service demand. Modeling studies suggest that targeted early interventions based on risk prediction can significantly reduce long-term healthcare costs and productivity losses [64]. Therefore, the integration of a Long COVID risk assessment into national care strategies can enhance individual care pathways to provide a critical tool for managing the wider public health and economic consequences of the pandemic.

## 5. Conclusions

A structured risk assessment strategy for Long COVID is vital for enabling early intervention and optimizing healthcare resource allocation. The dose–response relationship we established between spike protein concentration and disease symptoms, including proinflammatory effects, can be used as a predictive tool for Long COVID risk assessment. The time-dependent spike protein modeling can incorporate the most sensitive proinflammatory mediator (CXCL8) as a biomarker for preventive measures. By quantifying spike protein levels in patient samples and correlating them with symptom severity and inflammatory biomarker profiles, clinicians can identify individuals at higher risk for persistent symptoms. This relationship allows for the stratification of patients into risk categories, facilitating personalized monitoring and early therapeutic interventions. Our work provides a mechanistic basis for targeting treatments aimed at reducing spike protein load or mitigating their downstream inflammatory effects, potentially preventing or alleviating Long COVID manifestations. Additionally, to strengthen preparedness for further outbreaks of post-viral conditions, future studies can build on the standardized risk assessment research and include other crucial mediators for patients suffering from Long COVID.

## Figures and Tables

**Figure 1 viruses-17-01215-f001:**
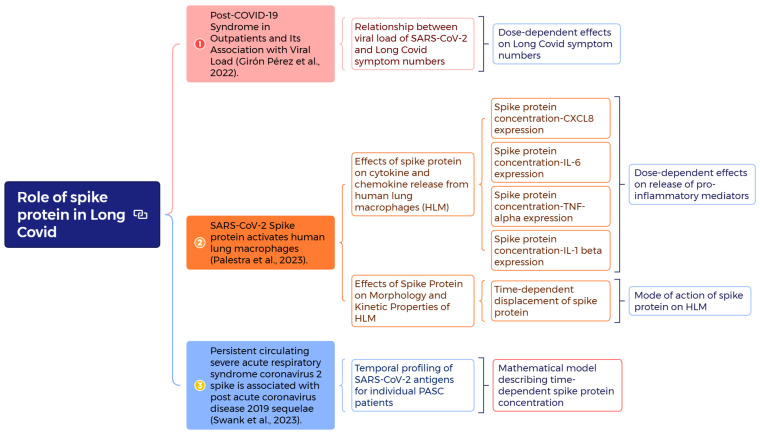
Schematic presents a mind mapping (https://xmind.ai/ (accessed on 5 July 2025)) to investigate the role of spike proteins in Long COVID with analyses of dose–response effects on symptom numbers [22] and proinflammatory mediators [24], the mode of action of spike proteins on Human Lung Macrophages (HLM) [24], and the construction of a mathematical model to describe time-dependent spike protein concentration [23].

**Figure 2 viruses-17-01215-f002:**
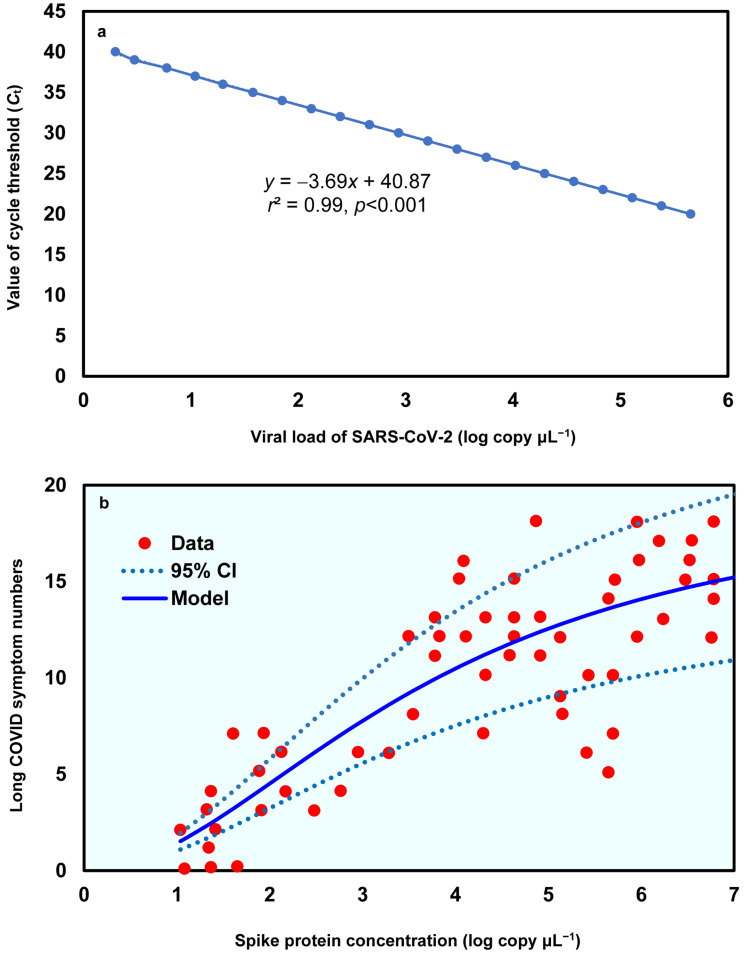
(**a**) Linear relationship between value of cycle threshold (*C*_t_) and the viral load of SARS-CoV-2 in Gene E and (**b**) the dose–response relationship between spike protein concentration (log copy μL^−1^) and Long COVID symptom numbers.

**Figure 3 viruses-17-01215-f003:**
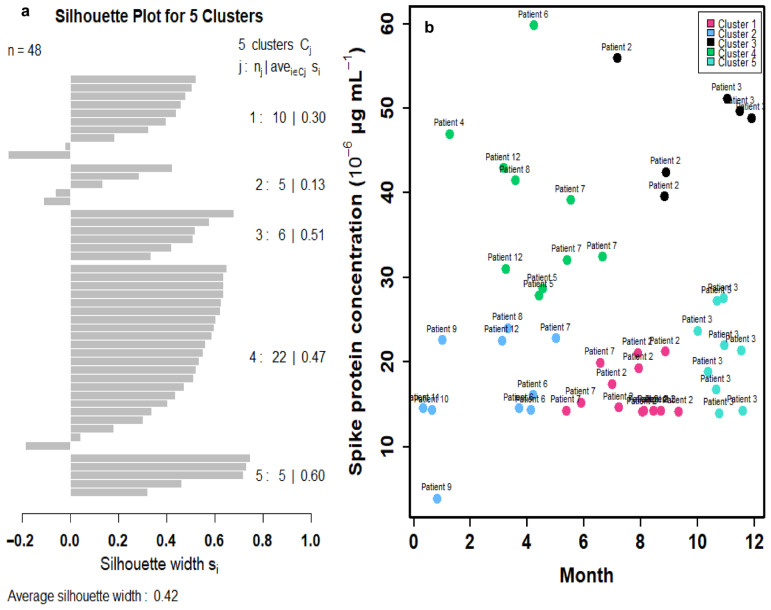
(**a**) Silhouette plot depicting the quality of cluster assignments based on spike protein concentration levels in 12 Long COVID patients with 48 data points. Each bar represents a patient’s silhouette coefficient, reflecting how similar they are to their own cluster compared to other clusters. Higher silhouette values indicate better-defined and more cohesive clustering, suggesting distinct patterns in spike protein concentration among patient subgroups. (**b**) Cluster analysis of spike protein concentrations in 12 Long COVID patients identified five distinct clusters based on silhouette scores.

**Figure 4 viruses-17-01215-f004:**
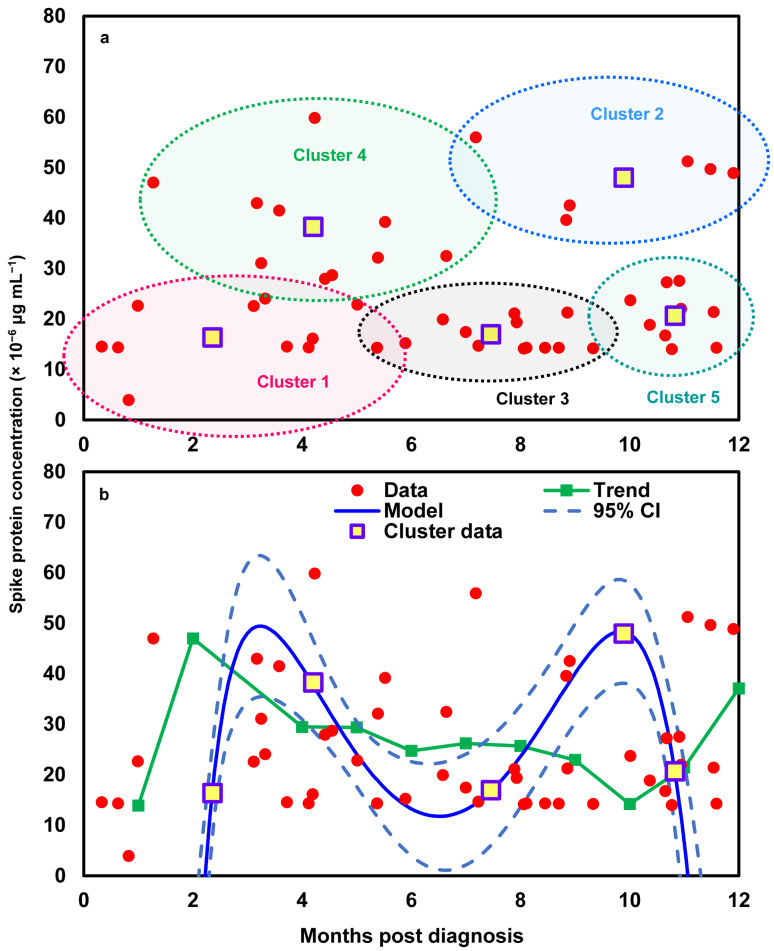
(**a**) Visualization of individual data points and cluster centroids representing the average spike protein concentration within each of the five identified clusters and (**b**) the development of a mathematical model describing time-dependent spike protein concentrations based on the average values of the five identified clusters, compared with the trend derived from monthly average concentrations across the 12 Long COVID patients.

**Figure 5 viruses-17-01215-f005:**
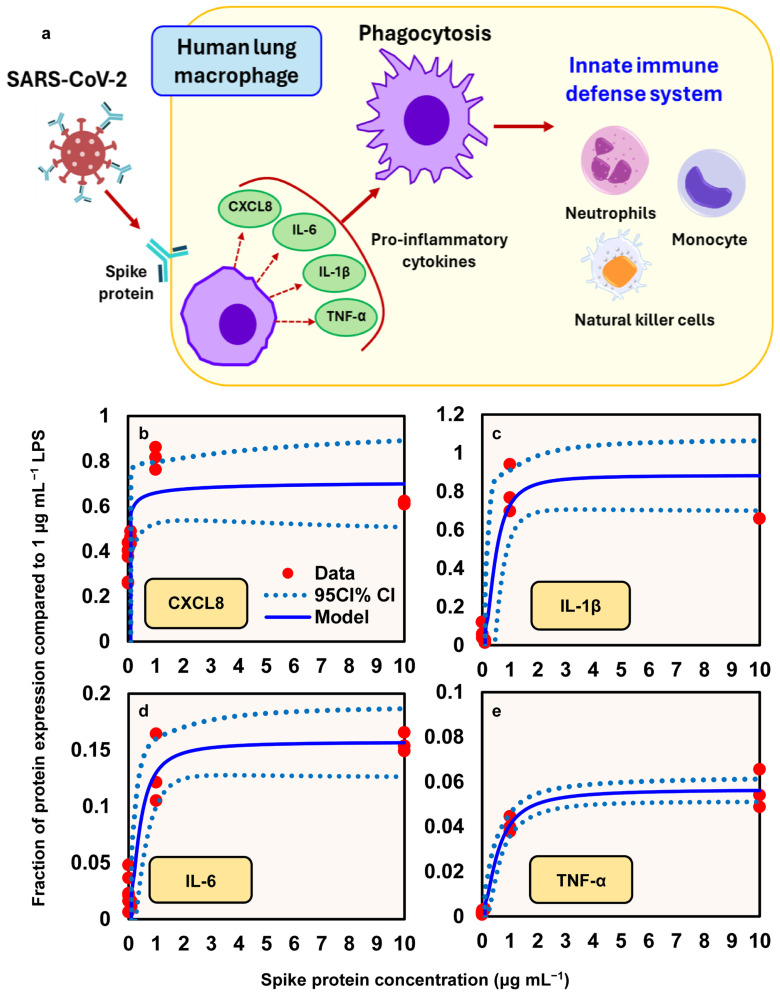
(**a**) Illustration of the mechanism by which spike proteins induce the release of proinflammatory cytokines (e.g., IL-6, IL-1β, TNF-α) and the chemokine CXCL8 from human lung macrophages (HLMs), promoting phagocytosis and contributing to innate immune responses. Hill models describing the relationship between spike protein concentration (µg/mL) and the relative expression of proinflammatory mediators, expressed as a fraction of the response to 1 µg/mL lipopolysaccharide (LPS): (**b**) CXCL8; (**c**) IL-1β; (**d**) IL-6; (**e**) TNF-α.

## Data Availability

No new data were generated for this review.

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
