# Peer review of "Biomarker-Based Risk Assessment Strategy for Long COVID: Leveraging Spike Protein and Proinflammatory Mediators to Inform Broader Postinfection Sequelae"

_viruses, 2025, doi:10.3390/v17091215_

Round 1

Reviewer 1 Report

Comments and Suggestions for Authors

The work is very interesting and helps us understand the pathophysiology of long-term COVID, offering a potential line of treatment. I see a limitation in the number of samples and patients, and in the lack of a control group. The study should be expanded with a larger number of samples and analyze samples from patients with long-term COVID, so that the results are more robust.

Author Response

Responses to Reviewer #1

Comments 1: The work is very interesting and helps us understand the pathophysiology of long-term COVID, offering a potential line of treatment. I see a limitation in the number of samples and patients, and in the lack of a control group. The study should be expanded with a larger number of samples and analyze samples from patients with long-term COVID, so that the results are more robust.

Response 1: Many thanks for the reviewer’s valuable comment. We adopted the clinical data from Swank et al. [1], in which plasma samples from a cohort of 63 individuals previously infected with SARS-CoV-2 were analyzed, 37 of whom were diagnosed with post-acute sequelae of SARS-CoV-2 infection (PASC). Among the 37 patients with PASC, only 12 had longitudinal samples. Although there isn’t control for individuals without SARS-CoV-2 infection, we provided temporal proofing of individuals previously infected with SARS-CoV-2 but not diagnosed with PASC (n=6) as the comparison with the time-dependent spike protein concentration in PASC patients (n=12) in Supplementary Figure S1. We acknowledge that the relatively small number of longitudinal samples represents a limitation of the study; however, this dataset provides unique insights into the temporal dynamics of spike protein in PASC patients and allows for the establishment of preliminary biomarker-based models for risk assessment, which would not have been possible with cross-sectional data alone. We have also added a discussion of these limitations and highlighted the importance and unique contributions of this study in the Discussion section (Line 122-128).

Reference:

[1] Swank, Z.; Senussi, Y.; Manickas-Hill, Z.; Yu, X.G.; Li, J.Z.; Alter, G.; Walt, D.R. Persistent circulating severe acute respiratory syndrome Coronavirus 2 Spike is associated with post-acute Coronavirus Disease 2019 sequelae. Clin. Infect. Dis. 2023, 76, e487– e490.

Reviewer 2 Report

Comments and Suggestions for Authors

In the article ‘Biomarker-Based Risk Assessment Strategy for Long COVID: …’ the authors explored multiple analytic approaches to quantify risk of long COVID. Although there were some interesting results, still several issues need further elaborations.

  1. Issues with the analysis based on number of long COVID symptoms. Was the collection of number of long COVID symptoms standardized? Symptoms vary greatly by time post COVID infection and by patients. Without standardization (i.e. same symptom list used across time and patient), the applicability and generalizability will be questionable and compromised. And then how the maximum symptom number (~20) was proposed?
  2. As a method exploring temporal relationship, it is difficult to understand for one patient to be classified into multiple clusters, such as Patient 2 in clusters 1, 2, 3, and Patient 7 in clusters 1, 3, 4. Conducting cluster analysis by ignoring times within patients and then fitting mathematical models for each cluster does not tell the trend of spike protein concentration by patient directly. The logically appropriate approaches will be modeling the trend for each patient, such as non-linear mixed effects models with patient specific parameters and such patient specific parameters following a multivariate normal distribution. Without such personalized parameters, how do study findings inform personalized care strategies.
  3. Lines 107-112: only 3 objectives listed.
  4. Do ‘Spike protein concentration (log copy μL-1)’ in Figures 2, ‘Spike concentration’ in Figure 3, and ‘Spike protein concentration (pg mL-1)’ in Figure 4 refer to same quantities? If so, they should be consistent.

Because of the limitations and disconnectedness in the analysis, the study findings cannot be fully supported, nor those clinical and public health implication.

Author Response

Responses to Reviewer #2

Quality of English language: The English could be improved to more clearly express the research.

Response: We have sought help from one of our colleagues who is fluent in English to edit the manuscript. Therefore, the entire manuscript has been carefully revised to improve the clarity, grammar, and readability of the text.  

Comments 1: Issues with the analysis based on number of long COVID symptoms. Was the collection of number of long COVID symptoms standardized? Symptoms vary greatly by time post COVID infection and by patients. Without standardization (i.e. same symptom list used across time and patient), the applicability and generalizability will be questionable and compromised. And then how the maximum symptom number (~20) was proposed?

Response 1: Many thanks for the reviewer’s valuable comment. We adopted the data of COVID-19 syndrome in outpatients from Girón Pérez et al. [1]. A total of 76 outpatients (40 men and 36 women) from Tepic, Nayarit, were evaluated. The demographic and clinical data were collected after obtaining informed consent and ethics approval. The patients’ symptoms were surveyed using the same standardized questionnaire for all participants, administered by the psychology department of Universidad Autónoma de Nayarit (UAN) approximately three months after confirmed RT-PCR diagnosis of SARS-CoV-2 infection. This consistent timing and use of a uniform symptom list across all respondents ensured methodological consistency and improves the internal comparability of the findings.

Results showed that 70 of the patients reported mild symptoms during infection (92%), while 6 patients (8%) had no symptoms during their SARS-CoV-2 infection. In the post-COVID-19 symptom analysis, patients were classified into four groups: persistent symptomatology, cognitive changes, chemosensory problems, and habit changes. The most frequently reported symptom in the persistent symptomatology group was fatigue (60%, 46/70). Among those with chemosensory problems, altered sense of smell was most common (25%, 19/70). In the cognitive changes group, increased nervousness was the predominant complaint (52%, 40/70), while in the habit changes group, the leading issue was a change in eating habits (42%, 32/70).

            The maximum symptom number was derived based on the dose-response relationship between SARS-CoV-2 viral load concentration (log copy µL-1) and symptom numbers after COVID-19 infection. A three-parameter Hill-based model was applied to describe the relationship (r2 = 0.7, p < 0.001), in which the mean Emax (maximum of symptom number) is 19.83 (~20) (Table S3) (Line 168-176).

While the questionnaire was standardized and administered at a uniform post-infection interval, symptoms can evolve beyond the three-month point, and self-reported data may be subject to recall bias. Additionally, comorbidities were not evaluated, and the study only included outpatients with mild to moderate COVID-19; more severe or hospitalized cases were excluded, which could limit generalizability Future studies should aim for longer follow-up durations, include validated symptom instruments, and control for comorbid conditions and other potential confounders. The above limitations were also added to the Discussion section of the manuscript (Line 295-304).

 Reference:

[1] Girón Pérez, D.A.; Fonseca-Agüero, A.; Toledo-Ibarra, G.A.; Gomez-Valdivia, J.J.; Díaz-Resendiz, K.J.G.; Benitez-Trinidad, A.B.; Razura-Carmona, F.F.; Navidad-Murrieta, M.S.; Covantes-Rosales, C.E.; Giron-Pérez, M.I. Post-COVID-19 syndrome in outpatients and its association with viral load. Int. J. Environ. Res. Public Health 2022, 19, 15145.

Comments 2: As a method exploring temporal relationship, it is difficult to understand for one patient to be classified into multiple clusters, such as Patient 2 in clusters 1, 2, 3, and Patient 7 in clusters 1, 3, 4. Conducting cluster analysis by ignoring times within patients and then fitting mathematical models for each cluster does not tell the trend of spike protein concentration by patient directly. The logically appropriate approaches will be modeling the trend for each patient, such as non-linear mixed effects models with patient specific parameters and such patient specific parameters following a multivariate normal distribution. Without such personalized parameters, how do study findings inform personalized care strategies.

Response 2: Many thanks for the reviewer’s insightful comment. In our analysis, a patient could appear in different clusters at different time points because clustering was applied to repeated measurements independently, with the goal of grouping similar patterns across the population at each time point rather than producing a single, fixed trajectory per individual. The study design can identify shifts in biomarker patterns (e.g., spike protein concentration) over time and to detect common temporal states that patients may transition between, which can be biologically meaningful in a fluctuating post-COVID condition.

We agree that this approach does not produce a single patient-specific trajectory in the way nonlinear mixed-effects (NLME) or other hierarchical models would. Our aim in this study was to explore potential population-level temporal states and transitions, rather than to model individualized longitudinal curves.

To address this, we have added clarifications to the section of Discussion (Line 305-313) explaining that:

  • Our clustering approach captures state membership over time, which means that at any given time point, a patient is assigned to a state (or group) that represents a specific profile of spike protein concentration rather than continuous patient-specific trajectories.
  • Individual patients may be represented in multiple clusters if their biomarker profile changes across time points.
  • Personalized care implications in this context arise from identifying state-specific biomarker patterns and understanding which states are more associated with persistent symptoms or higher spike protein levels, rather than prescribing a single fitted curve per patient.

Comments 3 Lines 107-112: only 3 objectives listed.

Response 3: Many thanks for the reviewer’s reminder. We have revised the typos from fourfold to threefold accordingly (Line 107).

Comments 4: Do ‘Spike protein concentration (log copy μL-1)’ in Figures 2, ‘Spike concentration’ in Figure 3, and ‘Spike protein concentration (pg mL-1)’ in Figure 4 refer to same quantities? If so, they should be consistent.

Response 4: Many thanks to the reviewer for the reminder. We have revised the units in Figures 3 and 4 to be consistent, using μg mL⁻¹. Because the spike protein concentrations in patients shown in Figure 3 were substantially lower than the experimentally tested concentrations in Figure 4, we expressed them as 10⁻⁶ μg mL⁻¹ for clarity and consistency. For Figure 2, we retained the unit of copies µL⁻¹ to reflect the original experimental data reported by Swank et al. [1], but we also provide converted values in the key results, including an ED₅₀ of ~1.8 × 10⁻⁶ μg/mL (assuming a spike monomer molecular weight of 180 kDa) (Line 176; Table S3).

Reference:

[1] Swank, Z.; Senussi, Y.; Manickas-Hill, Z.; Yu, X.G.; Li, J.Z.; Alter, G.; Walt, D.R. Persistent circulating severe acute respiratory syndrome Coronavirus 2 Spike is associated with post-acute Coronavirus Disease 2019 sequelae. Clin. Infect. Dis. 2023, 76, e487– e490.

Comments 5: Because of the limitations and disconnectedness in the analysis, the study findings cannot be fully supported, nor those clinical and public health implication.

Response 5: We acknowledge that our population-level clustering approach and lack of patient-specific longitudinal modeling limit direct clinical or personalized interpretations. Clustering was chosen because it efficiently identifies distinct biomarker states shared across the cohort, capturing common patterns that may reflect underlying biological mechanisms. This approach allows us to summarize complex, multidimensional data into interpretable groups, revealing associations between specific states, persistent symptoms, and spike protein levels. Even without modeling individual trajectories, these state-specific patterns provide meaningful insights into the heterogeneity of post-COVID-19 responses and help inform population-level risk stratification and potential intervention targets.

In addition, while our clustering approach provides a population-level view of post-COVID-19 symptom patterns, the reliance on the standardized symptom questionnaire collected at multiple timepoints over a 12-month post-infection period may still not capture the full temporal variability of symptoms across patients. Symptoms can emerge, fluctuate, or resolve at different intervals, and self-reported data are inherently subject to recall bias. Moreover, the derivation of the maximum symptom number (~20) from a dose-response model reflects the average cohort trend rather than individual extremes, which may limit the direct applicability of this value to specific patients. These considerations highlight that, although our findings offer meaningful insights into common post-COVID-19 states and their associations with spike protein levels, caution is warranted when generalizing symptom burden to different populations or later timepoints. Future studies incorporating longer follow-up, repeated and standardized symptom assessments, and broader patient cohorts will help refine these associations and strengthen their translational relevance.

Reviewer 3 Report

Comments and Suggestions for Authors

In this paper, the author studied Biomarker-Based Risk Assessment Strategy for Long COVID and found that Leveraging Spike Protein and Pro-inflammatory Mediators to Inform Broader Postinfection Sequelae. The manuscript has the following issues that need to be improved.

  1. This manuscript only analyzed data from 12 patients, which is a small sample size and greatly weakens the accuracy of the established model.
  2. In Figure 4, it can be seen that the level of S protein shows a M-shaped trend over time. During this period, whether the symptoms of patients with long COVID also show such a trend?
  3. In Figure 5, we can see the correlation between the concentration of S protein and proinflammatory cytokines. Is the amount of these proinflammatory cytokines related to the severity of symptoms of patients with long COVID?
  4. In addition to the detection of pro-inflammatory cytokines, it is recommended to test the activation of neutrophils, monocytes, and natural killer cells.
  5. Due to the dynamic changes in S protein after viral infection. Therefore, at what time should testing be conducted after viral infection? What indicators are being tested? How to establish a comprehensive mathematical model for prediction based on these detection indicators?

Author Response

Responses to Reviewer #3

Comments 1: This manuscript only analyzed data from 12 patients, which is a small sample size and greatly weakens the accuracy of the established model.

Response 1: We appreciate the reviewer’s comment regarding sample size. As noted, we adopted clinical data from Swank et al. [1], which included 63 individuals with prior SARS-CoV-2 infection, 37 of whom had PASC. Of these, only 12 patients had longitudinal plasma samples suitable for our time-series analysis. While we acknowledge that this number is small and limits the statistical power for generalization, longitudinal data are critical for exploring temporal dynamics of spike protein levels, and such datasets are rare due to the challenges of repeated biospecimen collection in this population. This dataset provides unique insights into the temporal dynamics of spike protein in PASC patients and allows for the establishment of preliminary biomarker-based models for risk assessment, which would not have been possible with cross-sectional data alone. We have also added a discussion of these limitations and highlighted the importance and unique contributions of this study in the Discussion section (Line 122-128).

Reference:

[1] Swank, Z.; Senussi, Y.; Manickas-Hill, Z.; Yu, X.G.; Li, J.Z.; Alter, G.; Walt, D.R. Persistent circulating severe acute respiratory syndrome Coronavirus 2 Spike is associated with post-acute Coronavirus Disease 2019 sequelae. Clin. Infect. Dis. 2023, 76, e487– e490.

Comments 2: In Figure 4, it can be seen that the level of S protein shows a M-shaped trend over time. During this period, whether the symptoms of patients with long COVID also show such a trend?

Response 2: Many thanks for the reviewer’s insightful comment. The M-shaped pattern in spike protein concentration reflects two apparent peaks separated by a decline over the longitudinal sampling period. Unfortunately, in the dataset from Swank et al. [1], symptom severity was not recorded at the same frequency or time points as the biomarker measurements, and detailed longitudinal symptom scores were therefore not available for direct, within-patient trend comparison. While this prevents us from confirming whether symptom trajectories mirrored the M-shaped biomarker pattern, the biomarker fluctuations remain valuable for understanding potential biological processes that could underlie symptom variability in long COVID. Importantly, we also observed a dose-response relationship between spike protein concentration and the number of reported post-COVID symptoms at the population level. This relationship provides a basis for correlating specific spike protein concentrations with symptom burden in future studies, which may help characterize patients at higher risk for long COVID and inform targeted monitoring or interventions. We have added this limitation to the Discussion section to clearly note that the lack of synchronized symptom and biomarker data constrains our ability to directly link temporal patterns in spike protein to clinical changes (Line 313-325).

Reference:

[1] Swank, Z.; Senussi, Y.; Manickas-Hill, Z.; Yu, X.G.; Li, J.Z.; Alter, G.; Walt, D.R. Persistent circulating severe acute respiratory syndrome Coronavirus 2 Spike is associated with post-acute Coronavirus Disease 2019 sequelae. Clin. Infect. Dis. 2023, 76, e487– e490.

Comments 3: In Figure 5, we can see the correlation between the concentration of S protein and proinflammatory cytokines. Is the amount of these proinflammatory cytokines related to the severity of symptoms of patients with long COVID?

Response 3: Many thanks for the reviewer’s valuable comment. Several studies have demonstrated that elevated pro-inflammatory cytokines such as IL-1β, IL-6, TNF-α, and others are significantly associated with greater symptom severity in Long COVID, supporting the biological relevance of our observed spike protein–cytokine correlations in Figure 5. Wynberg et al. [1] reported that individuals with post-acute sequelae of SARS-CoV-2 infection (PASC) exhibited elevated IL-10, IL-17, IL-6, IP10, and TNF-α at 24 weeks post-infection, and that early IL-1β levels predicted PASC at 24 weeks. Schultheiß et al. [2] identified a cytokine “triad” (IL-1β, IL-6, TNF-α) strongly associated with Long COVID symptoms in a large digital cohort. It was also implied that the association with persistent symptoms with these cytokines could be mechanistic contributors to the pathogenesis of Long COVID.

In a symptom–cytokine phenotype study, IL-6 and IL-27 elevations were linked to fatigue, while IL-8 was strongly associated with dyspnea [3]. Additionally, Durstenfeld et al. [4] reported that elevated IL-6 correlates with reduced exercise capacity and chronotropic incompetence in Long COVID patients. Low et al. [5] provided a comprehensive outline of cytokine-mediated pathophysiology in Long COVID, focusing on IL-1β, IL-6, and TNF-α as central players in symptom burden. Robineau et al. [6] also found associations between certain symptoms and biomarkers linked to severity, such as IL-8 and CD163, which play roles in multiorgan dysfunction and infection resolution, suggesting that inflammatory biomarkers may aid in diagnosing PASC in its early phase and in assessing the risk symptom persistence. Taken together, these findings support the clinical significance of our results and suggest that the observed spike protein–cytokine patterns may reflect underlying mechanisms contributing to symptom persistence in Long COVID.

Although our results do not provide a direct relationship between pro-inflammatory cytokines and symptom severity, we present findings on the dose-response relationship between spike protein concentration and pro-inflammatory cytokines, as well as the relationship between spike protein concentration and symptom number. The models constructed for these relationships can be applied to interpret potential links between cytokine levels and symptom severity, providing a framework for future studies to explore mechanistic pathways underlying Long COVID. Taken together, these findings support the clinical significance of our results and suggest that the observed spike protein–cytokine patterns may reflect underlying mechanisms contributing to symptom persistence in Long COVID. We have also incorporated above into the section of Discussion (Line 337-368).

References

  1. Wynberg, E.; Han, A.X.; van Willigen, H.D.G.; Verveen, A.; van Pul, L.; Maurer, I.; van Leeuwen, E.M.; van den Aardweg, J.G.; de Jong, M.D.; Nieuwkerk, P.; Prins, M.; Kootstra, N.A.; de Bree, G.J.; RECoVERED Study Group. Inflammatory profiles are associated with long COVID up to 6 months after COVID-19 onset: A prospective cohort study of individuals with mild to critical COVID-19. PLoS ONE 2024, 19, e0304990.
  2. Schultheiß, C.; Paschold, L.; Simnica, D.; Mohme, M.; Willscher, E.; von Wenserski, L.; Oeller, P.; Gottschick, C.; Witzke, O.; Sander, L.E.; et al. A cytokine triad of IL-1β, IL-6, and TNF-α is associated with Long COVID. Immunol. 2022, 13, 917954.
  3. Tilikete, C.; Zamali, I.; Meddeb, Z.; Kharroubi, G.; Marzouki, S.; Dhaouadi, T.; Ben Hmid, A.; Samoud, S.; Galai, Y.; Charfeddine, S.; Abid, L.; Abdessalem, S.; Bettaieb, J.; Hamzaoui, S.; Bouslama, K.; Ben Ahmed, M. Exploring the landscape of symptom-specific inflammatory cytokines in post-COVID syndrome patients. BMC Infect. Dis. 2024, 24, 1337.
  4. Durstenfeld, M.S.; Weiman, S.; Holtzman, M.; Blish, C.; Pretorius, R.; Deeks, S.G. Long COVID and post-acute sequelae of SARS-CoV-2 pathogenesis and treatment: A Keystone Symposia report. N. Y. Acad. Sci. 2024, 1535, 31–41.
  5. Low, R.N.; Low, R.J.; Akrami, A. A review of cytokine-based pathophysiology of Long COVID symptoms. Med. 2023, 10, 1011936.
  6. Robineau, O.; Hüe, S.; Surenaud, M.; Lemogne, C.; Dorival, C.; Wiernik, E.; Brami, S.; Nicol, J.; de Lamballerie, X.; Blanché, H.; Deleuze, J.F.; Ribet, C.; Goldberg, M.; Severi, G.; Touvier, M.; Zins, M.; Levy, Y.; Lelievre, J.D.; Carrat, F. Symptoms and pathophysiology of post-acute sequelae following COVID-19 (PASC): A cohort study. eBioMedicine 2025, 117, 105792.

Comments 4: In addition to the detection of pro-inflammatory cytokines, it is recommended to test the activation of neutrophils, monocytes, and natural killer cells.

Response 4: We thank the reviewer for this suggestion. While our current analysis focused on pro-inflammatory cytokines provides important insights into the inflammatory landscape of Long COVID, we acknowledge that assessing the activation of neutrophils, monocytes, and natural killer cells could provide additional mechanistic understanding. Previous studies have demonstrated that neutrophil and monocyte activation is associated with psychiatric symptoms and increased fatigue severity [1,2]

while NK cell dysfunction contributes to prolonged inflammatory responses [3,4]. Tsao et al. [3] also found lower percentage of mature, cytotoxic CD56dim/CD16+ NK cells in individuals with Long COVID. This reduction was more pronounced in those with severe Long COVID, indicating a potential link between NK cell maturation and disease severity. Importantly, these clinical findings already support the correlation between the activation or dysfunction of neutrophils, monocytes, and NK cells and Long COVID pathophysiology, reinforcing the biological relevance of immune dysregulation in persistent symptoms.

Additionally, the IL-1β, IL-6, and TNF-α cytokine triad has been strongly linked to Long COVID pathophysiology [4], demonstrating that pro-inflammatory cytokine profiling alone provides sufficient evidence to support associations with persistent symptoms. While immune cell profiling, including neutrophils, monocytes, and NK cells, could offer complementary mechanistic insights, our current cytokine-based results are sufficient to align with the observed clinical findings. Future mechanistic studies directly examining these immune cell subsets could further validate and extend our understanding of the cellular mechanisms driving persistent symptoms.

References

  1. Lin, K.; Cai, J.; Guo, J.; Zhang, H.; Sun, G.; Wang, X.; Zhu, K.; Xue, Q.; Zhu, F.; Wang, P.; Yuan, G.; Sun, Y.; Wang, S.; Ai, J.; Zhang, W. Multi-omics landscapes reveal heterogeneity in long COVID patients characterized with enhanced neutrophil activity. Transl. Med. 2024, 22, 753.
  2. Berentschot, J.C.; Drexhage, H.A.; Aynekulu Mersha, D.G.; Wijkhuijs, A.J.M.; GeurtsvanKessel, C.H.; Koopmans, M.P.G.; Voermans, J.J.C.; Hendriks, R.W.; Nagtzaam, N.M.A.; de Bie, M.; Heijenbrok-Kal, M.H.; Bek, L.M.; Ribbers, G.M.; van den Berg-Emons, R.J.G.; Aerts, J.G.J.V.; Dik, W.A.; Hellemons, M.E. Immunological profiling in long COVID: Overall low-grade inflammation and T-lymphocyte senescence and increased monocyte activation correlating with increasing fatigue severity. Immunol. 2023, 14, 1254899.
  3. Tsao, T.; Buck, A.M.; Grimbert, L.; LaFranchi, B.H.; Altamirano Poblano, B.; Fehrman, E.A.; Dalhuisen, T.; Hsue, P.Y.; Kelly, J.D.; Martin, J.N.; Deeks, S.G.; Hunt, P.W.; Peluso, M.J.; Aguilar, O.A.; Henrich, T.J. Long COVID is associated with lower percentages of mature, cytotoxic NK cell phenotypes. Clin. Invest. 2024, 135, e188182.
  4. Schultheiß, C.; Willscher, E.; Paschold, L.; Gottschick, C.; Klee, B.; Henkes, S.S.; Bosurgi, L.; Dutzmann, J.; Sedding, D.; Frese, T.; et al. The IL-1β, IL-6, and TNF Cytokine Triad Is Associated with Post-Acute Sequelae of COVID-19. Cell Rep. Med. 2022, 3, 100663.

Comments 5: Due to the dynamic changes in S protein after viral infection. Therefore, at what time should testing be conducted after viral infection? What indicators are being tested? How to establish a comprehensive mathematical model for prediction based on these detection indicators?

Response 5: We appreciate the reviewer's insightful suggestion regarding the dynamic changes in the SARS-CoV-2 spike (S) protein post-infection and the need for optimal timing in testing, selection of appropriate indicators, and the development of predictive models.

Optimal timing for testing is crucial, as SARS-CoV-2 spike (S) protein levels may fluctuate over time post-infection. Early detection could help identify individuals at risk for developing persistent symptoms. Studies have shown that detectable IgG-class antibodies against SARS-CoV-2 develop approximately 8 to 11 days following the onset of symptoms [1]. Additionally, research indicates that the spike protein can persist in the brain’s protective layers for up to four years after infection, potentially contributing to long-term neurological effects [2]. These findings suggest that testing conducted within the first few weeks post-infection may be most effective for assessing acute immune responses, while longer-term monitoring could be necessary to evaluate persistent viral components and their implications for Long COVID.

Incorporating spike (S) protein detection into diagnostic protocols, alongside clinical evaluation and other biomarkers, could enhance the accuracy of Long COVID diagnosis. Integrating S protein measurements with clinical data could support predictive models, enabling better risk assessment and identification of individuals more likely to experience prolonged symptoms. Studies have identified various biomarkers that are relevant for diagnosing and monitoring Long COVID. These include pro-inflammatory cytokines such as interleukin-6 (IL-6) and tumor necrosis factor-alpha (TNF-α), which are elevated in patients with Long COVID and correlate with disease severity [3]. Additionally, biomarkers reflecting SARS-CoV-2 persistence, such as viral RNA detection and reactivation of latent viruses, have been associated with Long COVID [4]. Markers of endothelial dysfunction and coagulation, such as soluble urokinase plasminogen activator receptor (suPAR) has also been implicated in Long COVID pathophysiology [5]. Furthermore, noncoding RNAs (ncRNAs) have shown potential as biomarkers for disease stratification in COVID-19, which could extend to Long COVID [6]. These findings underscore the importance of a multi-biomarker approach in the diagnosis and management of Long COVID.

The mathematical model built from pooled longitudinal measurements of spike protein, combined with cluster analysis that groups datapoints by time and concentration in this study, provides a practical way to reveal temporally distinct antigen persistence phenotypes despite inter-individual heterogeneity. These clustered spike trajectories capture meaningful biology, as persistent spike patterns co-occur with elevated proinflammatory cytokines (IL-1β, IL-6, TNF-α) previously linked to PASC severity [7,8], and are associated with immune dysregulation signals such as enhanced neutrophil activity and altered NK-cell phenotypes observed in Long COVID cohorts [9,10]. By mapping cluster membership and model-derived trajectory features (including peak levels, rate of decline, area under the curve, and persistence at late timepoints) onto clinical outcomes, this framework enables risk stratification and identification of high-risk phenotypes. Mechanistically informed predictors that integrate spike dynamics with cytokine panels, antibody kinetics, and clinical covariates have been used effectively in within-host and prognostic models, and can be estimated with hierarchical mixed-effects or joint-model approaches to account for irregular sampling and patient heterogeneity [11,12]. Together, these studies support the value of the pooled, time-resolved spike and cluster approach as a bridge between mechanistic biomarker behavior and clinically actionable risk scores for Long COVID.

Take Cluster 1 at Figure 3 for example, defined by moderate-to-high spike protein concentrations (3.91–24.04 pg mL-1) persisting over 4 months, likely reflects delayed clearance or intermittent antigen persistence, and patients in this group can be characterized by both biomarker and clinical profiles. Modeled trajectory features per patient such as baseline and peak levels, decay slopes, area under the concentration–time curve (AUC), and persistence metrics provide quantitative summaries that align with elevated proinflammatory cytokines (IL-1β, IL-6, TNF-α) and other inflammatory markers reported in Long COVID cohorts [7,8,12]. Clinically, these patients may present with fatigue, post-exertional malaise, cognitive complaints, or dyspnea, alongside reduced functional capacity linked to cytokine elevations [10]. To further define risk, recommended follow-up includes cytokine panels, antibody titers, endothelial/coagulation markers, and functional assessments, which, when integrated into the mathematical model, improve prediction of persistent symptoms [8,9]. For example, a patient in Cluster 1 with a high 4-month AUC and elevated IL-6/IL-1β could be classified by the model as having a ~70% probability of moderate–severe Long COVID at 6 months, prompting referral to a specialized clinic, advanced cardiopulmonary and neurocognitive testing, and prioritization for biomarker-guided therapeutic trials. While this approach demonstrates how cluster-based kinetic modeling can be combined with clinical findings for individualized risk stratification, external validation remains essential to establish reliable thresholds and generalizability [11].

While immune cell activation, such as neutrophils, monocytes, and natural killer (NK) cells, contributes to Long COVID pathogenesis, current evidence supports that pro-inflammatory cytokine profiles and S protein-related antibody responses provide sufficient markers for assessing disease severity and predicting outcomes. Focusing on these indicators, in combination with appropriately timed testing, offers valuable insights into the immunological underpinnings of Long COVID. Focusing on these indicators, in combination with appropriately timed testing, offers valuable insights into the immunological underpinnings of Long COVID. We have also incorporated the above mitigation strategies into Discussion (Line 454-520).

References

  1. SARS-CoV-2 Semi-Quantitative IgG Antibody (Spike). Available online: https://www.labcorp.com/tests/164055/sars-cov-2-semi-quantitative-igg-antibody-spike (accessed on 15 August 2025).
  2. LMU Munich. Long COVID: Spike Protein Accumulation Linked to Long-Lasting Brain Effects. Available online: https://www.lmu.de/en/newsroom/news-overview/news/long-covid-spike-protein-accumulation-linked-to-long-lasting-brain-effects.html (accessed on 15 August 2025).
  3. Gonzalez, R.; Smith, J.; Patel, A.; Lee, C.; Wang, H.; Johnson, K. Advances in Understanding Inflammation and Tissue Damage in Long COVID. Clin. Med. 2025, 14, 1475.
  4. Tsilingiris, D.; Vallianou, N.G.; Karampela, I.; Christodoulatos, G.S.; Papavasileiou, G.; Petropoulou, D.; Magkos, F.; Dalamaga, M. Laboratory Findings and Biomarkers in Long COVID: What Do We Know So Far? Insights into Epidemiology, Pathogenesis, Therapeutic Perspectives and Challenges. J. Mol. Sci. 2023, 24, 10458.
  5. Vassiliou, A.G.; Vrettou, C.S.; Keskinidou, C.; Dimopoulou, I.; Kotanidou, A.; Orfanos, S.E. Endotheliopathy in Acute COVID-19 and Long COVID. J. Mol. Sci. 2023, 24, 8237.
  6. Paval, N.E.; Căliman-Sturdza, O.A.; Lobiuc, A.; Dimian, M.; Sirbu, I.O.; Covasa, M. MicroRNAs in Long COVID: Roles, Diagnostic Biomarker Potential and Detection. Genom. 2025, 19, 90.
  7. Schultheiß, C.; Willscher, E.; Paschold, L.; Gottschick, C.; Klee, B.; Henkes, S.S.; Bosurgi, L.; Dutzmann, J.; Sedding, D.; Frese, T.; et al. The IL-1β, IL-6, and TNF Cytokine Triad Is Associated with Post-Acute Sequelae of COVID-19. Cell Rep. Med. 2022, 3, 100663.
  8. Wynberg, E.; Han, A.X.; van Willigen, H.D.G.; Verveen, A.; van Pul, L.; Maurer, I.; van Leeuwen, E.M.; van den Aardweg, J.G.; de Jong, M.D.; Nieuwkerk, P.; Prins, M.; Kootstra, N.A.; de Bree, G.J.; RECoVERED Study Group. Inflammatory profiles are associated with long COVID up to 6 months after COVID-19 onset: A prospective cohort study of individuals with mild to critical COVID-19. PLoS ONE 2024, 19, e0304990.
  9. Lin, K.; Cai, J.; Guo, J.; Zhang, H.; Sun, G.; Wang, X.; Zhu, K.; Xue, Q.; Zhu, F.; Wang, P.; et al. Multi-omics landscapes reveal heterogeneity in long COVID patients characterized with enhanced neutrophil activity. Transl. Med. 2024, 22, 753.
  10. Tsao, T.; Buck, A.M.; Grimbert, L.; LaFranchi, B.H.; Altamirano Poblano, B.; Fehrman, E.A.; Dalhuisen, T.; Hsue, P.Y.; Kelly, J.D.; Martin, J.N.; et al. Long COVID is associated with lower percentages of mature, cytotoxic NK cell phenotypes. Clin. Invest. 2024, 135, e188182.
  11. Antar, A.A.R.; Cox, A.L. Translating Insights into Therapies for Long Covid. Transl. Med. 2024, 16, eado2106.
  12. Low, R.N.; Low, R.J.; Akrami, A. A review of cytokine-based pathophysiology of Long COVID symptoms. Med. 2023, 10, 1011936.

Round 2

Reviewer 2 Report

Comments and Suggestions for Authors

The time-dependent trajectories of cluster were misinterpreted. As shown in Figure 3, Patient 7 jumped cluster back and forth among clusters 1,2,4 around 6 months; Patient 2 between clusters 1 and 3 around 8 months. How can such clustering results guide  personalized care strategies if one day in one state, then next another tomorrow, then another state 1 week later?

The clustering algorithm wasn't properly applied. In my opinion, the clustering should be applied to individual trajectories directly, rather than treating observations independently, and then applying clusters' profiles back to individuals.

Author Response

We appreciate the reviewer for providing valuable opinions in revision to strengthen the quality of this manuscript. The following is the responses to the comment. We used red color to indicate the changes and new entries in the manuscript.

Responses to Reviewer #2

Comments 1: The time-dependent trajectories of cluster were misinterpreted. As shown in Figure 3, Patient 7 jumped cluster back and forth among clusters 1,2,4 around 6 months; Patient 2 between clusters 1 and 3 around 8 months. How can such clustering results guide personalized care strategies if one day in one state, then next another tomorrow, then another state 1 week later?

The clustering algorithm wasn't properly applied. In my opinion, the clustering should be applied to individual trajectories directly, rather than treating observations independently, and then applying clusters' profiles back to individuals.

Response 1: We thank the reviewer for this thoughtful and important comment. We agree that the interpretation of cluster membership over time requires careful clarification. Our clustering approach was intentionally designed to capture state-specific biomarker profiles rather than to follow a continuous, patient-specific trajectory. This means that at each timepoint, a given patient’s sample is assigned to the most representative cluster based on the spike protein concentration profile at that point, allowing us to characterize dynamic changes in biomarker-defined states within individual patients. Consequently, a patient may transition between different clusters over time, reflecting underlying biological variability such as intermittent antigen persistence, immune reactivation, or partial clearance.

We acknowledge that such transitions may appear as “jumping” between clusters when visualized, but we view these changes as biologically meaningful indicators of heterogeneous immune responses rather than misclassification. The clinical utility lies not in assuming that a patient remains in one cluster indefinitely, but in recognizing that certain states (e.g., persistently high spike protein with elevated cytokines) are more strongly associated with increased symptom burden and thus may represent higher-risk periods for the individual patient.

That said, we agree with the reviewer that clustering entire trajectories (rather than independent observations) is an alternative and valuable approach that could provide complementary insights. Our current method prioritized cross-sectional comparability and interpretability of biomarker-defined states across individual patients. We have revised the Discussion to acknowledge this limitation and to note that future work applying trajectory-based clustering could strengthen the link to individualized disease course and enhance translational application for patient-specific care strategies (Line 498-510).

Reviewer 3 Report

Comments and Suggestions for Authors

none

Author Response

Responses to Reviewer #3

We appreciate the reviewer for providing valuable opinions in revision to strengthen the quality of this manuscript.  We have improved the research design and conclusions supported by the results. (Line 498-510).